# Immobilized Whole-Cell Transaminase Biocatalysts for Continuous-Flow Kinetic Resolution of Amines

**Zsófia Molnár** [1,2,3], **Emese Farkas** [1], **Ágnes Lakó** [1], **Balázs Erdélyi** [2], **Wolfgang Kroutil** [4], **Beáta G. Vértessy** [3,5], **Csaba Paizs** [6] **and László Poppe** [1,6,7,*]

[1] Department of Organic Chemistry and Technology, Budapest University of Technology and Economics, Műegyetem rkp. 3, 1111 Budapest, Hungary; molnar.zsofia@mail.bme.hu (Z.M.); farkas.emese@mail.bme.hu (E.F.); lako.agnes@mail.bme.hu (Á.L.)

[2] Fermentia Microbiological Ltd., Berlini út 47–49, 1405 Budapest, Hungary; balazs.erdelyi@fermentia.hu

[3] Institute of Enzymology, Research Center for Natural Sciences, Hungarian Academy of Science, Magyar tudósok krt. 2, 1117 Budapest, Hungary; vertessy@mail.bme.hu

[4] Institute of Chemistry, University of Graz, NAWI Graz, BioTechMed Graz, BioHealth, Heinrichstraße 28, 8010 Graz, Austria; wolfgang.kroutil@uni-graz.at

[5] Department of Applied Biotechnology and Food Science, Budapest University of Technology and Economics, Műegyetem rkp. 3, 1111 Budapest, Hungary

[6] Biocatalysis and Biotransformation Research Centre, Faculty of Chemistry and Chemical Engineering, Babeş-Bolyai University of Cluj-Napoca, Arany János Str. 11, RO-400028 Cluj-Napoca, Romania; paizs@chem.ubbcluj.ro

[7] SynBiocat Ltd., Szilasliget u 3, H-1172 Budapest, Hungary

\* Correspondence: poppe@mail.bme.hu; Tel.: +36(1)463-3299

**Abstract:** Immobilization of transaminases creates promising biocatalysts for production of chiral amines in batch or continuous-flow mode reactions. *E. coli* cells containing overexpressed transaminases of various selectivities and hollow silica microspheres as supporting agent were immobilized by an improved sol-gel process to produce immobilized transaminase biocatalysts with suitable stability and mechanical properties for continuous-flow applications. The immobilized cell-based transaminase biocatalyst proved to be durable and easy-to-use in kinetic resolution of four racemic amines **1a–d**. The batch and continuous-flow mode kinetic resolutions with transaminase biocatalyst of opposite stereopreference provided access to both enantiomers of the corresponding amines. By using the most suitable immobilized transaminase biocatalysts, this study describes the first transaminase-based approach for the production of both pure enantiomers of 1-(3,4-dimethoxyphenyl)ethan-1-amine **1d**.

**Keywords:** stereoselective biocatalysis; transaminase; kinetic resolution; flow chemistry; sol-gel; whole-cell immobilization

## 1. Introduction

Enantiopure amines are essential chiral building blocks for the synthesis of a wide variety of active pharmaceutical ingredients. Chemical synthesis of these compounds usually employs transition metal catalysts of relatively high toxicity, and may require harsh reaction conditions. In recent years, there has been a growing interest in transaminases (TAs), which offer a sustainable alternative to these synthetic chemical processes [1–3]. Transaminases have been successfully used in the preparation of several pharmaceutically relevant compounds, like 3-aryl-γ-aminobutyric acid derivatives [4], sitagliptin [5] and valinol [6].

Transaminases can be applied for separation of the enantiomers from their racemates in kinetic resolution (KR) with a maximum of 50% yield, or for conversion of a prochiral carbonyl compound to a

chiral amine in asymmetric synthesis with a theoretical yield of 100%. Although asymmetric synthesis could provide higher yields, it usually suffers from disfavored reaction equilibrium [7]. Furthermore, the costs of enzyme production indicates the need for biocatalyst recovery in an economically viable process [8]. These are the most significant difficulties hindering the widespread industrial application of transaminases. Several strategies have been developed to overcome the difficulties arising from the disfavored reaction equilibrium in asymmetric synthesis with TAs, such as the removal of products by extraction [9], by evaporation [10] or by coupled cascade reactions [11,12]. The overall efficiency of the KR process may be enhanced by recycling the formed ketone to racemic amine by a proper reductive amination method [13]. Although these solutions work well in batch reactions, for the sustainable, industrial production of enantiopure amines the intensification possibilities offered by immobilized TA biocatalysts and by the continuous-mode operations are needed [1]. This trend is indicated by the successful applications of TAs in continuous-flow reactors which have been developed in the past years [14–23].

A wide range of immobilization methods for transaminases have been reported over the last couple of years. Transaminases have been successfully immobilized by using different kinds of carriers, such as polymeric resins [7,16,23], functionalized cellulose [21], chitosan [24–26], inorganic-based nanoflowers [27,28], macrocellular silica monoliths [19], $MnO_2$ nanorods [29] or porous glass metal affinity supports [22]. In addition to immobilization of TAs in their isolated and fully or partially purified form, the entrapment of cell-free extracts, or even whole cells with comprising TA activity in different sol-gel matrices [14,30,31] proved to be an excellent approach to prepare high-performance and stable immobilized biocatalysts. In many cases, utilization of whole cells over enzyme solutions is advantageous due to the lower production costs, increased stability and easier handling [32]. Immobilized transaminases have been successfully employed in the preparation of several enantiopure compounds, such as 1-phenoxypropane-2-amine [33], 3-amino-1-Boc-piperidine [34], 1-methyl-3-phenylpropylamine [23,26] and mexiletine [16].

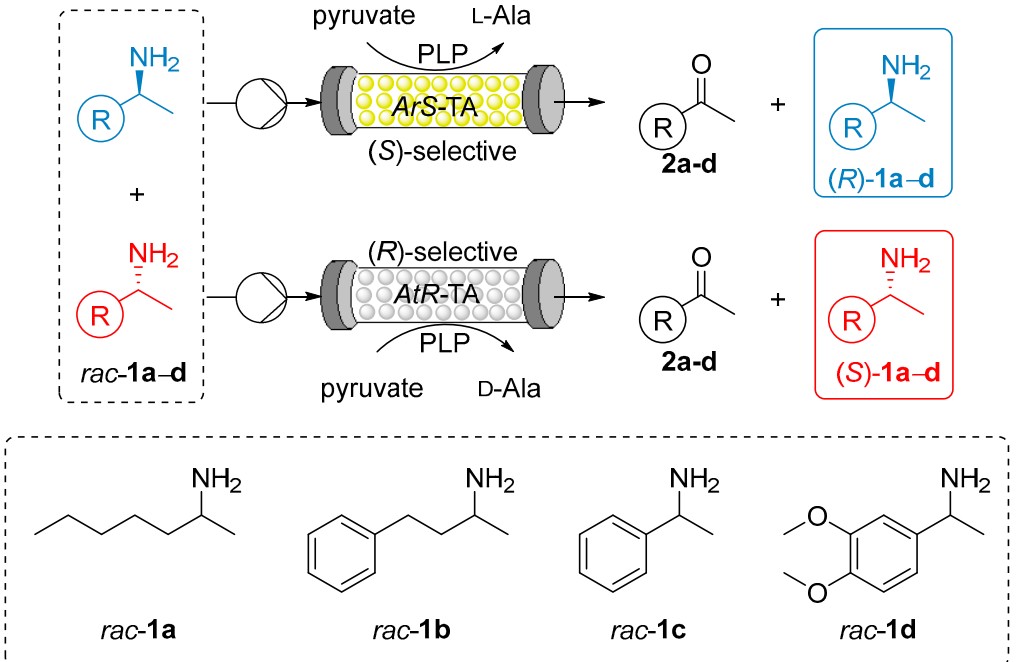

**Scheme 1.** Continuous-flow kinetic resolution of primary amines *rac*-**1a–d** with immobilized whole-cell transaminase biocatalysts.

This study aimed at the preparation of robust, stereoselective TA biocatalysts capable of operating under continuous-flow conditions to produce enantiopure amines. A promising sol-gel-based whole-cell immobilization method has already been applied in a previous work of our group for

the transaminase from *Chromobacterium violaceum* expressed in *E. coli* [14]. Hereby we report the immobilization of further transaminases with different enantioselectivity and substrate specificity. The immobilized cells proved to be easy-to store, cheap and durable biocatalysts, and were applied successfully in the continuous kinetic resolution of racemic amines (Scheme 1).

## 2. Results and Discussion

### 2.1. Immobilized Recombinant Whole-Cells as Transaminase Biocatalysts

In this study, six different transaminase-expressing *E. coli* whole-cells were immobilized by an improved sol-gel entrapment method including three (*S*)-selective TAs [from *Vibrio fluvialis* (*VfS*-TA), *Arthrobacter* sp. (*ArS*-TA) and a mutated variant of *Chromobacterium violaceum* TA (*CvS*-TA$_m$)] and three (*R*)-selective TAs [from *Aspergillus terreus* (*AtR*-TA), *Arthrobacter sp.* (*ArR*-TA) and its mutated variant (*ArR*-TA$_m$)].

A multistep immobilization protocol was used to prepare the whole-cell TA biocatalysts (Figure 1). First, a silica sol was formed by the acid-catalyzed hydrolysis of tetraethyl orthosilicate (TEOS). After the sol formation, a suspension consisting of *E. coli* cells containing the recombinant transaminase and hollow silica microspheres as supporting aid was added to the sol. The silica microspheres possess excellent properties for the adsorption of whole cells due to their high specific surface area grafted with aminoalkyl and vinyl functions. Upon standing the formed mixture of the cells and microspheres in silica sol, gelation occurred. Aging of the formed sol-gel was performed at 4 °C for 2 days. Finally, the formed sol-gel powder was gently crushed and washed with water. The immobilized whole-cell TA biocatalyst were then dried at room temperature for 1 day, then stored at 4 °C in the form of powder.

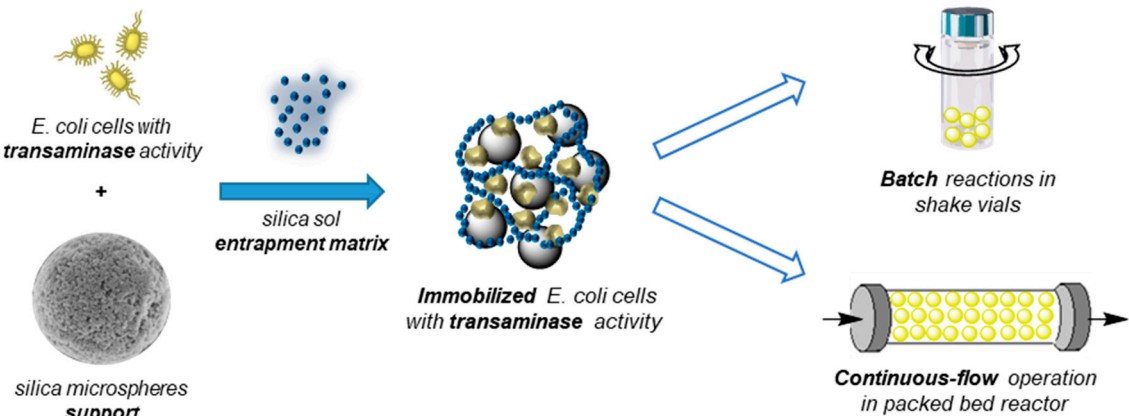

**Figure 1.** Process for immobilization of transaminase-containing recombinant *E. coli* cells. The immobilized transaminase (TA) biocatalysts can be applied both in batch (e.g., shake vials) and in continuous-flow operations (e.g., packed bed reactors).

The immobilization method applied in this study combines the advantages of cell-adsorption and sol-gel entrapment. The silica microspheres provide the good mechanical properties of the biocatalyst, while the entrapping silica matrix afford the high immobilization yield (~100% of the cells were retained [14]; ~0.9 g of dry TA biocatalyst could be produced from 1 g of wet cells). The immobilization could be scaled up from g scale to 10 g scale without any noticeable problem. The immobilized TA-containing whole-cells proved to be easy to produce, reproducible and robust biocatalysts operating remarkably well in continuous-flow systems. Despite of the apparently harsh conditions during the immobilization (strong acidic environment, drying at atmospheric pressure and room temperature), the transaminase containing cells remained catalytically active in their air-dried, well-preserved immobilized form for many months. Activity yield could be determined by comparison of the activity data with the wet cells vs. with the immobilized TA biocatalysts. At the optimum pH (~7.5) activity yields in the range of 80–105% could be observed for the six immobilized TA biocatalysts (see Section 3

and Figure S18 in the Supplementary Materials). The immobilized biocatalysts retained 95% activity after storing for 6 months in refrigerator (determined by testing the kinetic resolution of *rac*-**1c** at 24 h reaction time).

*2.2. Kinetic Resolution of Amines with Immobilized Transaminases*

2.2.1. Shake Vial Screening of Immobilized Transaminases in Kinetic Resolution of Amines *rac*-**1a–d**

The immobilized TA biocatalysts were tested in the kinetic resolution of amines *rac*-**1a–d** with an aim to choose the best performing (*S*)- and the best performing (*R*)-selective transaminase biocatalyst for further testing in continuous-flow mode KRs. While amines *rac*-**1a–c** were frequently used as model substrates for transaminases, to our best knowledge the KR of *rac*-**1d** has never been tested with transaminases before. The only examples of biocatalytic approaches to the enantiomers of **1d** are lipase-catalyzed kinetic resolution [35] or dynamic kinetic resolution [36]. The results of the screening experiments are shown in Table 1.

**Table 1.** Shake vial screening of immobilized TA biocatalysts in kinetic resolution of amines *rac*-**1a–d**.

| Entry | TA [a] | 1a [b] | | 1b [b] | | 1c [b] | | 1d [b] | |
|---|---|---|---|---|---|---|---|---|---|
| | | *c* (%) [c] | *ee* (%) [c] | *c* (%) [c] | *ee* (%) [c] | *c* (%) [c] | *ee* (%) [c] | *c* (%) [c] | *ee* (%) [c] |
| 1 | *ArS*-TA | 53 | 98.1 (*R*) | 51 | 89.6 (*R*) | 44 | 79.9 (*R*) | 51 | >99.8 (*R*) |
| 2 | *VfS*-TA | 50 | 64.5 (*R*) | 47 | 66.8 (*R*) | 39 | 65.0 (*R*) | 13 | 13.1 (*R*) |
| 3 | *CvS*-TA$_m$ | 38 | 39.3 (*R*) | 44 | 76.2 (*R*) | 17 | 21.1 (*R*) | 23 | 26.4 (*R*) |
| 4 | *AtR*-TA | 51 | 97.0 (*S*) | 48 | 86.4 (*S*) | 42 | 72.9 (*S*) | 51 | >99.8 (*S*) |
| 5 | *ArR*-TA | 57 | 99.7 (*S*) | 49 | 90.4 (*S*) | 35 | 53.7 (*S*) | 36 | 50.1 (*S*) |
| 6 | *ArR*-TA$_m$ | 47 | 97.5 (*S*) | 33 | 29.9 (*S*) | 21 | 27.3 (*S*) | 15 | 15.5 (*S*) |

[a] Source organism and former abbreviations of the transaminases: *ArS*-TA: *Arthrobacter* sp., (*S*)-selective (ArS-ωTA [37]); *VfS*-TA: *Vibrio fluvialis* (Vf-ωTA [38]); *CvS*-TA$_m$: *Chromobacterium violaceum*, mutant (Cv-ωTA$_{W60C}$ [23,39]); *ArR*-TA: *Arthrobacter* sp., (*R*)-selective (ArR-ωTA [40]); *AtR*-TA: *Aspergillus terreus* (AT-ωTA [40]); *ArR*-TA$_m$: *Arthrobacter* sp. (ArRmut11-ωTA [40]). [b] Reagents and conditions in 5 mL reaction: amine *rac*-**1a–d** (30 mM), immobilized transaminase (100 mg), sodium pyruvate (22.5 mM), pyridoxal-5′-phosphate monohydrate (PLP, 0.3 mM), phosphate buffer (pH 7.5, 100 mM), 30 °C, 24 h. [c] Conversion (*c*) of *rac*-**1a–d** to the corresponding ketone **2a–d** and enantiomeric excess (*ee*) for the residual amine (*R*)-**1a–d** or (*S*)-**1a–d** were determined after derivatization with Ac$_2$O by gas chromatography (GC) on chiral stationary phase.

The immobilized TA biocatalysts were effective for the KRs of the investigated amines *rac*-**1a–d** in batch reactions. Regarding the (*S*)-selective TAs, *CvS*-TA$_m$ had the lowest catalytic activity, *VfS*-TA and *ArS*-TA were able to catalyze the conversion of the amines to the corresponding ketone with conversions similar to each other. However, *ArS*-TA had better enantioselectivity for all of the investigated substrates than the other two (*S*)-selective TAs, with superior performance in the KR of *rac*-**1d**. Among the investigated (*R*)-selective TAs, *ArR*-TA$_m$ (the mutant variant of *ArR*-TA) had the lowest overall activity, since this variant was engineered for the bioconversion of large, bulky substrates [5]. *ArR*-TA and *AtR*-TA performed comparably in the KRs of the tested amines *rac*-**1a–d**. Nonetheless, in case of the KR of *rac*-**1d**, the activity of *AtR*-TA exceeded the one of *ArR*-TA. Although lower conversions (*c*) from *rac*-**1c–d** were reached with *ArR*-TA than the corresponding ones with *AtR*-TA, the values of enantiomeric ratio (*E*)—which could be calculated from the corresponding conversion (*c*)-enantiomeric excess (*ee*) values [41]—indicated highly enantiomer selective KRs. The performance of *CvS*-TA$_m$ could be evaluated similarly; the conversion (*c*)-enantiomeric excess (*ee*) values experienced in KRs from *rac*-**1b–d** indicated highly enantiomer selective transformations at lower conversion rates. Based on these results, the (*S*)-selective *ArS*-TA and the (*R*)-selective *AtR*-TA biocatalysts were selected for the further continuous-flow experiments.

### 2.2.2. Kinetic Resolution of Amines *rac*-**1a–d** with Immobilized TA Biocatalysts in Continuous-Flow Mode

The conditions for the kinetic resolution of amines **1a–d** in continuous-flow mode were optimized using *ArS*-TA and *AtR*-TA. First, the immobilized whole-cell TA biocatalyst-filled columns were fed with the racemic amine (either of the *rac*-**1a–d**)-containing solution at 7.5 mM substrate concentration by varying the flow rates from 40 to 100 µL min⁻¹. If even at the highest flow rate (100 µL min⁻¹; limited by achievable pressure) the 50% theoretical conversion was achieved while maintaining high enantiomeric excess (>98%), further experiments with increased substrate concentrations were performed. The best conditions found in this way are displayed in Table 2 for *ArS*-TA and in Table 3 for *AtR*-TA.

**Table 2.** Production of (*R*)-**1a–d** by kinetic resolution of *rac*-**1a–d** with whole-cell immobilized *ArS*-TA under optimized conditions in continuous-flow mode [a].

| Entry | Substrate | Substrate Conc. (mM) | Flow Rate (µL min⁻¹) | Conversion (%) | $ee_{(R)\text{-}1}$ (%) |
|---|---|---|---|---|---|
| 1 | *rac*-1a | 7.5 | 100 | 50 | 99.1 |
| 2 | *rac*-1b | 20 | 100 | 51 | >99.8 |
| 3 | *rac*-1c | 50 | 50 | 50 | >99.8 |
| 4 | *rac*-1d | 20 | 40 | 53 | 99.1 |

[a] For reaction details see Section 3.5.1.

**Table 3.** Production of (*S*)-**1a–d** by kinetic resolution of *rac*-**1a–d** with whole-cell immobilized *AtR*-TA under optimized conditions in continuous-flow mode [a].

| Entry | Substrate | Substrate Conc. (mM) | Flow Rate (µL min⁻¹) | Conversion (%) | $ee_{(S)\text{-}1}$ (%) |
|---|---|---|---|---|---|
| 1 | *rac*-1a | 7.5 | 100 | 52 | 99.2 |
| 2 | *rac*-1b | 7.5 | 50 | 51 | >99.8 |
| 3 | *rac*-1c | 7.5 | 50 | 49 | 98.8 |
| 4 | *rac*-1d | 10 | 60 [b] | 53.5 | 99.2 |

[a] For reaction details see Section 3.5.1. [b] Two columns connected serially.

Similar to the KRs performed in batch mode, the immobilized *ArS*-TA biocatalyst exhibited excellent activity and good to high enantiomer selectivity in continuous-flow mode KRs of the racemic amines *rac*-**1a–d**. While the KR of *rac*-**1a** with the *ArS*-TA biocatalyst could be performed with 50% conversion at the lowest selected substrate concentration (7.5 mM), full conversion of the (*S*)-enantiomers could be achieved at significantly higher substrate concentrations of *rac*-**1b** and *rac*-**1c** resulting in the (*R*)-**1b** and (*R*)-**1c** with excellent *ee* values. The enantiomer selectivity of the *ArS*-TA biocatalyst towards *rac*-**1d** was less pronounced, which was indicated by several runs with conversion values well over 50% indicating that significant amount of (*R*)-**1d** was also transformed. However, by raising the flow rate, which resulted in decreased residence time, a proper conversion of 53% from *rac*-**1d** could be achieved with good enantiomeric excess of the residual (*R*)-**1d** ($ee_{(R)\text{-}\mathbf{1d}}$ = 99.1%).

When the (*R*)-selective immobilized whole-cell *AtR*-TA was employed as biocatalyst, almost perfect kinetic resolution of amines *rac*-**1a–c** could be performed with full consumption of the (*S*)-enantiomer (Table 3: *c* = 49–52% from *rac*-**1a–c**); however, only at relatively low substrate concentrations (7.5 mM of *rac*-**1a–c**). In the case of *rac*-**1d** the conversion could not reach 50% in a single column even at the lowest investigated substrate concentration (7.5 mM of *rac*-**1d**); consequently, the complete KR of *rac*-**1d** had to be performed by means of two serially connected *AtR*-TA biocatalyst-filled columns. Similar to the *ArS*-TA, a conversion of *rac*-**1d** exceeding significantly the 50% (*R*)-enantiomer content was observed in the *AtR*-TA biocatalyst-catalyzed KRs at lower flow rates (≤50 µL min⁻¹), indicating a decreased degree of enantiomer selectivity as compared to the ones in KRs of the other three amines *rac*-**1a–c**. Nevertheless, increasing the flow rate of the continuous-flow KR of *rac*-**1d** to 60 µL min⁻¹ enabled a process with proper conversion (*c* = 53.5%) and enantiomeric excess ($ee_{(S)\text{-}\mathbf{1d}}$ = 99.2%), which was also suitable for performing preparative scale experiments.

### 2.2.3. Preparative Scale Kinetic Resolutions of *rac*-**1d** in Continuous-Flow Mode

To demonstrate the synthetic applicability of the selected novel immobilized whole-cell (*S*)- and (*R*)-selective TA biocatalysts, the KR of **1d** was performed on a preparative scale in continuous-flow mode (Figure 2). The pure enantiomers of the drug-like compound **1d** are potential building blocks for several reported drug candidates: Akincioglu [42] published a study of novel sulfamoyl carbamates and sulfamides as potential carbonic anhydrase and acetylcholine esterase inhibitors, while Kerr [43] studied the employment of *N*-(phenylpropyl)-1-arylethylamines as allosteric modulators of GABA$_B$ receptors with good results. These studies indicate the importance of efficient methods for production of the enantiomers amines like **1d**. The preparative scale continuous-flow KRs of *rac*-**1d** performed with the most efficient (*S*)-selective and (*R*)-selective immobilized whole-cell TA biocatalysts (*Ar*S-TA and *At*R-TA, respectively) resulted in efficient production of either (*S*)-**1d** or (*R*)-**1d** in good isolated yields and excellent enantiomeric excess (Table 4)

**Table 4.** Preparative scale production of either (*S*)-**1d** or (*R*)-**1d** by kinetic resolution of *rac*-**1d** in continuous-flow mode.

| Transaminase | Product | *ee* (%) | Yield (%) | Space Time Yield (mg cm$^{-3}$ day$^{-1}$) |
|---|---|---|---|---|
| *Ar*S-TA | (*R*)-1d [a] | 99.1 (*R*) | 45 | 115 |
| *At*R-TA | (*S*)-1d [b] | 99.2 (*S*) | 44 | 42.2 |

[a] 20 mM *rac*-**1d**, 40 µL min$^{-1}$. [b] 10 mM *rac*-**1d**, 60 µL min$^{-1}$ with two columns.

The system with immobilized *At*R-TA was stable for 21 h, producing virtually enantiopure (*S*)-**1d** ($ee_{(S)\text{-}1d}$ = 99.3%) with a space time yield of 42.2 mg cm$^{-3}$ day$^{-1}$ (Figure 2b). The immobilized *Ar*S-TA proved to be more durable, with more than 2 days of stable operation while producing apparently enantiopure (*R*)-**1d** ($ee_{(R)\text{-}1d}$ 99.1%) with an outstanding space time yield of 115 mg cm$^{-3}$ day$^{-1}$ (Figure 2a).

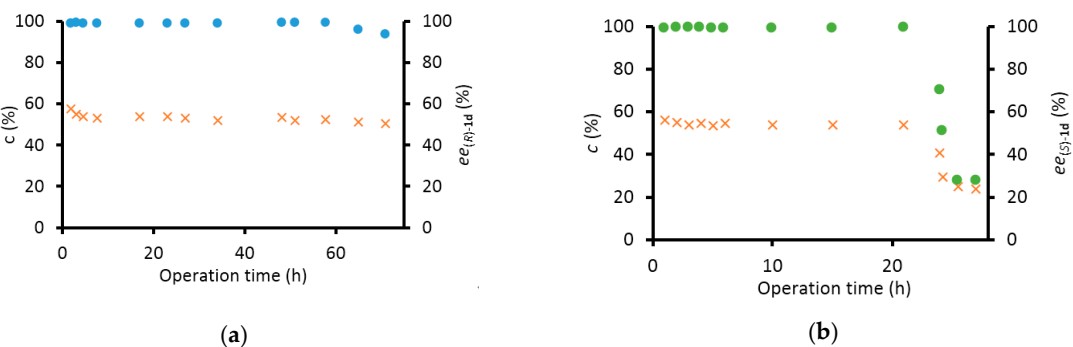

(**a**)    (**b**)

**Figure 2.** Long-term stability of the most efficient (*S*)-selective (*Ar*S-TA) and (*R*)-selective (*At*R-TA) immobilized TA biocatalysts in preparative scale continuous-flow kinetic resolution of *rac*-**1d** with (**a**) immobilized *Ar*S-TA; and (**b**) immobilized *At*R-TA [conversion (×), $ee_{(R)\text{-}1d}$ (●) and $ee_{(S)\text{-}1d}$ (●)].

Since the enzymes in this study were immobilized as whole-cell preparations, the transaminases themselves could not be stabilized by multipoint fixation to a support by this method. Several physiological processes could account to the observed loss of activity in the long term, e.g., the degradation of proteins, or the dissociation of the enzyme monomers from the heterodimeric structure inside the intracellular space.

## 3. Materials and Methods

### 3.1. Materials

Except where otherwise stated, all chemicals and starting materials were purchased from Sigma-Aldrich (St. Louis, MO, USA) or Alfa Aesar Europe (Karlsruhe, Germany). MAT540

(MATSPHERE™ SERIES 540-hollow silica microspheres etched with aminoalkyl and vinyl functions, with an average particle diameter of 10 μm) was purchased from Materium Innovations (Granby, QC, Canada). The procedure for the synthesis of *rac*-**1d** is described in the Supporting Materials (Section S2.1).

### 3.2. Recombinant Transaminase Production

Production of *ArS*-TA [37] and *VfS*-TA [38] was achieved in *E. coli* BL21(DE3) containing the recombinant pASK-IBA35+ plasmid with the gene of the given TA. LB-Car medium (5 mL; LB medium containing carbenicillin, 50 mg L$^{-1}$) was inoculated with one fresh colony from an overnight LB-Car agar plate and cells were grown overnight in shake flask (37 °C, at 200 rpm). LB medium (0.5 L) in a 2 L flask was inoculated with seed culture (2 mL) and cells were grown at 37 °C, 200 rpm until the OD$_{640}$ reached 0.8 (approx. 4 h). For induction, tetracycline solution (20 μL, 5 mg mL$^{-1}$ tetracycline in ethanol) was added and the culture was shaken for further 16 h at 25 °C, 200 rpm. The cells were then harvested by centrifugation (15,000 g, 4 °C, 20 min).

Production of *AtR*-TA [40], *ArR*-TA [40], *ArR*-TA$_m$ [40] and *CvS*-TA$_m$ [39] was achieved in *E. coli* BL21(DE3) containing the recombinant pET21a plasmid with the gene of the given TA. LB-Car medium (5 mL; LB medium containing carbenicillin, 50 mg L$^{-1}$) was inoculated with one fresh colony from an overnight LB-Car agar plate and cells were grown overnight in shake flask (37 °C, at 200 rpm). Autoinduction medium (0.5 L: Na$_2$HPO$_4$, 6 g L$^{-1}$; KH$_2$PO$_4$, 3 g L$^{-1}$; tryptone, 20 g L$^{-1}$; yeast extract, 5 g L$^{-1}$; NaCl, 5 g L$^{-1}$; glycerol, 7.56 g L$^{-1}$; glucose, 0.5 g L$^{-1}$; lactose, 2 g L$^{-1}$ [44]) in a 2 L flask was inoculated with seed culture (2 mL) and was shaken for 16 h at 25 °C, 200 rpm. The cells were then harvested by centrifugation (15,000× *g*, 4 °C, 20 min).

### 3.3. Immobilization of Transaminase-Expressing Whole-Cells

First a silica sol was prepared by addition of TEOS (14.4 mL) to a solution containing 0.1 M HNO$_3$ (1.3 mL) and distilled water (5 mL) followed by sonication of the resulted mixture for 5 min at room temperature (Emag Emmi 20HC Ultrasonic Bath, 45 kHz) and keeping the mixture at 4 °C for 24 h. Next, a cell paste suspension (10 g of paste of the centrifuged TA-expressing *E. coli* cells, suspended in 30 mL of 0.1 M phosphate buffer, pH 7.5) was mixed with a suspension of MAT540 support (3 g, suspended in 30 mL of 0.1 M phosphate buffer, pH 7.5) and shaken intensively until become homogeneous (Technokartell Test Tube Shaker Model T3SK, 40 Hz, room temperature, 5 min). Finally, the homogenized supported cell suspension was mixed with the silica sol and the resulted mixture was shaken intensively (Technokartell Test Tube Shaker Model T3SK, 40 Hz, room temperature, 5 min). Gelation occurred within 30 min at room temperature, followed by aging the gel at 4 °C for 48 h in an open dish. The crude immobilized TA biocatalyst was washed with distilled water (2 × 15 mL, 100 mM, pH 7.5), dried at room temperature (24 h), and stored at 4 °C to yield ~9 g of immobilized TA biocatalyst. The immobilized whole-cell TA biocatalysts could be stored in the refrigerator for months without significant loss of the original TA activity.

### 3.4. Kinetic Resolution of Amines rac-1a–d with the Immobilized TA Biocatalysts in Batch Mode

The immobilized TA biocatalyst (100 mg) was suspended in phosphate buffer (5 mL, 100 mM, pH 7.5) containing the racemic amine (*rac*-**1a–d,** 30 mM), sodium pyruvate (22.5 mM) and pyridoxal-5′-phosphate monohydrate (PLP, 0.3 mM) in 20 mL vials. The reaction mixture was shaken on an orbital shaker (600 rpm) at 30 °C for 24 h. To the samples taken from the reaction mixture (100 μL) sodium hydroxide (100 μL, 1 M) was added, followed by extraction with ethyl acetate (800 μL). Derivatization of the amines was performed by the addition of acetic anhydride (10 μL, 60 °C, 1 h), then the organic phase was dried over Na$_2$SO$_4$. Samples were analyzed by gas chromatography {analysis was performed on Agilent 4890 equipment with FID detector and Hydrodex β-6TBDM column [25 m × 0.25 mm × 0.25 μm film with heptakis-(2,3-di-*O*-methyl-6-*O*-*t*-butyldimethylsilyl)-β-cyclodextrine; Macherey & Nagel] using H$_2$ carrier gas (injector: 250 °C, detector: 250 °C, head pressure: 12 psi, split ratio: 50:1)}. For details see Supplementary Materials Table S1.

*3.5. Biotransformations with the Immobilized TA Biocatalysts in Continuous-Flow Mode*

The kinetic resolutions of *rac*-**1a–d** with the novel immobilized TA biocatalysts in continuous-flow mode were performed in a laboratory scale flow reactor built from a Knauer Azura P4.1S isocratic HPLC pump attached to SynBioCart™ columns (SynBiocat, Budapest, Hungary; stainless steel outer and PTFE inner tube, inner diameter: 4 mm; total length: 70 mm; packed length: 65 mm; inner volume: 0.816 mL) filled with the immobilized whole-cell biocatalyst (*ArS*-TA or *AtR*-TA biocatalyst, filling weights: 375 ± 12 mg a column) in an in-house made aluminum metal block column holder with precise temperature control. The columns were sealed by filter membranes made of PTFE [Whatman® Sigma-Aldrich, WHA10411311, pore size 0.45 μm]. The sealing elements were made of PTFE.

3.5.1. Kinetic Resolution of Racemic Amines *rac*-1**a–d** with Immobilized TA Biocatalysts in Continuous-Flow Mode

Kinetic resolutions of *rac*-**1a–d** in continuous-flow mode were performed in SynBioCart™ columns filled with the immobilized TA biocatalysts (*ArS*-TA or *AtR*-TA biocatalyst). The racemic amine (*rac*-**1a–d**; 7.5–50 mM) was dissolved in phosphate buffer (20 mL, 100 mM, pH 7.5) containing sodium pyruvate (0.75 equivalent to the racemic amine) and PLP (0.3 mM). The biocatalyst-filled column was pre-washed by phosphate buffer (50 μL min$^{-1}$ for 60 min) followed by pumping the substrate solution through the column at various flow rates (ranging from 40 μL min$^{-1}$ to 100 μL min$^{-1}$) at 30 °C. Samples were taken in triplicate after reaching the stationary phase, and analyzed (after extraction and derivatization as described above) by GC as described earlier. The standard deviation (SD) of measurements in triplicate did not exceed 1.5% of the mean value; therefore, SD values were not indicated in the Tables.

3.5.2. Preparative Scale Production of (*S*)- and (*R*)-1-(3,4-Dimethoxyphenyl)ethan-1-Amine [(*S*)-**1d** and (*R*)-**1d**] by Kinetic Resolution with Immobilized TA Biocatalysts in Continuous-Flow Mode

Kinetic resolutions of *rac*-**1d** in continuous-flow mode for preparative purposes were performed as described above operating the immobilized TA biocatalyst-filled (*ArS*-TA or *AtR*-TA biocatalyst) SynBioCart™ columns for longer periods. The residual amine [(*S*)-**1d** or (*R*)-**1d**] was isolated from the effluent during stationary phase of the operation. The pH of the collected effluent was adjusted to 1 by cc. HCl, and the forming ketone **2d** was removed by extraction with dichloromethane (3 × 40 mL). After removal of the ketone **2d**, the pH of the aqueous phase was adjusted to 10 by addition of ammonium hydroxide (25%) and the residual amine [(*S*)-**1d** or (*R*)-**1d**] was extracted with dichloromethane (3 × 40 mL). The unified organic phases were extracted with brine (30 mL), dried over Na$_2$SO$_4$ and concentrated in vacuum to yield the product amine [(*R*)-**1d**: 45% yield (227.1 mg, $ee_{(R)\text{-}\mathbf{1d}}$ 99.1%, $[\alpha]_D^{27}$ = +21.5 (c 1, EtOH)) from a 58 h long run at 40 μL min$^{-1}$ flow rate; or (*S*)-**1d**: 44% yield (60.3 mg, $ee_{(S)\text{-}\mathbf{1d}}$ 99.2%, $[\alpha]_D^{27}$ = −21.5 (c 1, EtOH)) from a 21 h long run at 60 μL min$^{-1}$ flow rate)].

## 4. Conclusions

In this study, immobilization of six transaminases—involving three (*S*)-selective and three (*R*)-selective transaminases—was investigated in their recombinant *E. coli* whole-cell forms together with hollow silica microspheres as support by entrapment in a sol-gel system. The immobilized whole-cell TA biocatalysts were able to catalyze the kinetic resolution of various amines efficiently in batch and in continuous-flow mode. The easy-to-use immobilized TA biocatalysts showed enhanced storage-stability and proved to be stable even in long-term continuous processes. To the best of our knowledge, this process is the first example for the transaminase-based production of the pure enantiomers of the drug-like 1-(3,4-dimethoxyphenyl)ethan-1-amine (*S*)-**1d** and (*R*)-**1d**. Our results demonstrate how continuous-flow operations can contribute to sustainable production of essential chiral building blocks with the aid of biocatalysis.

**Supplementary Materials:** The following are available online at http://www.mdpi.com/2073-4344/9/5/438/s1, Figure S1: $^1$H-NMR spectrum of *rac*-**1d**, Figure S2: $^{13}$C-NMR spectrum of *rac*-**1d**, Figure S3: FT-IR spectrum of *rac*-**1d**, Figure S4: GC chromatogram of *rac*-**1a** and **2a**, Figure S5: GC chromatogram of *rac*-**1b** and **2b**, Figure S6: GC chromatogram of *rac*-**1c** and **2c**, Figure S7: GC chromatogram of *rac*-**1d** and **2d**, Figure S8: GC chromatogram of KR from *rac*-**1d**, Figure S9: GC chromatogram of *(R)*-**1d** after working up the KR reaction mixture, Figure S10: $^1$H-NMR spectrum of *(R)*-**1d**, Figure S11: $^{13}$C-NMR spectrum of *(R)*-**1d**, Figure S12: FT-IR spectrum of *(R)*-**1d**., Figure S13: GC chromatogram of KR from *rac*-**1d**, Figure S14: GC chromatogram of *(S)*-**1d** after working up the KR reaction mixture, Figure S15: $^1$H-NMR spectrum of *(R)*-**1d**, Figure S16: $^{13}$C-NMR spectrum of *(R)*-**1d**, Figure S17: FT-IR spectrum of *(S)*-**1d**, Figure S18: pH dependence of the transaminase containing wet *E. coli* cells and the immobilized TA biocatalysts, Table S1: GC data of reference substrates and products.

**Author Contributions:** Conceptualization, Z.M., E.F., C.P. and L.P.; resources, B.E., W.K., B.G.V., C.P. and L.P.; writing—original draft preparation, Z.M., E.F. and Á.L.; writing—review and editing, B.E., B.G.V., W.K, C.P. and L.P.; supervision L.P.

**Funding:** This study was funded by Higher Education Excellence Program of the Ministry of Human Capacities in the frame of Biotechnology research area of Budapest University of Technology and Economics (BME FIKP-BIO) and further grants from the National Research, Development and Innovation Fund of Hungary (Budapest, Hungary; projects SNN-125637, K-119493, FIEK_16-1-2016-0007, VEKOP-2.3.2-16-2017-00013, 2017-1.3.1-VKE-2017-00013, 2018-1.2.1-NKP-2018-00005). Furthermore, we want to thank the financial support from NEMSyB, ID P37_273, Cod MySMIS 103413 funded by the Romanian Ministry for European Funds, through the National Authority for Scientific Research and Innovation (ANCSI) and cofounded by the European Regional Development Fund, Competitiveness Operational Program 2014-2020 (POC), Priority axis 1, Action 1.1. The authors also acknowledge the Gedeon Richter Talentum Foundation for the financial support, including PhD fellowship of Z. Molnár.

**Acknowledgments:** The authors are thankful to József Nagy and Gábor Hornyánszky (Budapest University of Technology and Economics, Hungary) for the helpful advices which greatly assisted the research.

**Conflicts of Interest:** The authors declare no conflict of interest.

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
