# Peer review of "Immobilized Whole-Cell Transaminase Biocatalysts for Continuous-Flow Kinetic Resolution of Amines"

_catalysts, doi:10.3390/catal9050438_

Round 1
Reviewer 1 Report
The manuscript by Molnár et al. describes the usage of immobilized E. coli cells heterologously expressing different (R)- and (S)-selective ω-transaminases (ω-TA). Immobilization methods on silica microspheres allow for reusage of the cells. With this system, kinetic resolution of four substrates has been tested in batch reactions and in a continous-flow operation in a packed bed reactor.
Results show good conversion rates and excellent enantiomeric access for the synthesis of both enantiomers of the tested compounds.
However, before recommending publication, especially the method section needs to be improved and complemented with the following revisions:
1. Enzymes
- Strains/ω-TA: Please add the PDB identity (or comparable data source) of the enzymes used in this study.
- What have been the criteria for choosing these very enzymes?
- What is meant by „mutated variant“ (line 83)? What has been mutated in CvS-TAm and ArR-Tam, and for what purpose?
- What is known about the expression levels of the enzymes in E. coli? Since Arthrobacter is Gram-positive and Aspergillus even eukaryotic – has some codon usage adaptation been carried out?
- Since W.K. is listed in the author contributions as „resource“, and one fourth [!] of all cited references stems from this group, it is likely that creation of the mutated enzymes has been performed in W.K.‘s lab and more information on the enzyme can be found in the cited literature – however, please specify.
2. Immobilization
- Please shift line 86-95 to the method section, maybe also 100-107.
- What is the average size/diameter of the silica microspheres?
- Experiments are adjusted by „100mg immobilized biocatalyst“ (line 252) which I think means „100mg silica microspheres with immobilized cells after drying“ – is that correct? Is it possible to calculate how many cells (or what cell dry weight) is immobilized per particle? Or activity per area or so?
- Can you comment on potential loss of activity by immobilization of the cells, e.g. by harsh immobilization conditions as well as by transport/diffusion limitation of substrate molecules through the gel? A comparison before and after immobilization is missing.
- Line 106 „the transaminase containing cells remained catalytically active […] for many months.“ This can be interpreted with „20% remaining activity after 3 months“ as well as “90% after 12 months” - please specify.
3. Further remarks:
- Table 1+2: What is the explanation for the apparent boost in enantiomeric excess for 1b conversion (a.o.) by ArS-TA when performing continuous flow (>99.8%/Tab. 2 compared to 89.6% with shake vial/Tab. 1)?
- Table 5: Although mathematically correct, it is misleading to label space time yields in [kg/m-3d-1] when only milligram quantities are produced (see line 296: „45% yield = 227.1mg“). There is no indication that scale-up to cubic meter scale can easily be performed
Author Response
Plese find our esponses to your notes in the uplooaded PDF file.

Reviewer 2 Report
The manuscript reports the co-immobilization of E. coli whole cells expressing transaminases and hollow silica microspheres by a sol-gel process, and the succesful application of the biocatalyst to continuous-flow resolution of racemic amines. Additionally, the most promising biocatalysts were proven to be efficient for the transaminase-catalysed production of pure enantiomers of 1-(3,4-dimethoxyphenyl)ethan-1-amine 1d.
The topic and the obtained results could be of interest for researchers in the field of biocatalysis and drug synthesis. This works appears to be the continuation of a previous one, but a considerable amount of new work is presented, as well as some novel contributions (i.e. application of the transaminase biocatalysts to the production of a compound potentially useful in the pharmaceutical sector).
The introduction is clearly written and provides the basic information needed to put the work in context, although some aspects could have been more thoroughly explained. The methods are described in detail. The results are clearly presented and the discussion is sound. In general, the quality of the article is good, and could merit publication in Catalysts. However, some relevant aspects should be considered before final approval:
- Some additional details on immobilized transaminases and their applications to enantiomer synthesis could be included in the Introduction section
- Statistical treatment of the data should be included throughout the manuscript
- Stability of immobilized biocatalysts is a key factor, and it is mentioned only superficially in the manuscript. Data on biocatalysts stability should be included, considering different factors (i.e. temperature, flow conditions, ionic strength, presence of solvents...)
- Some preliminary estimation of the cost of the proposed system should be included. Also, the suitability for scaling up should be discussed.
Author Response

(The authors gave the same response as above.)

Round 2
Reviewer 2 Report
The authors have not included in the manuscript all the aspects mentioned in the initial reviewer's report, but I consider the explanations provided in the response letter as satisfactory. Therefore, I recommend publication of the article.